# New Application of the Commercially Available Dye Celestine Blue B as a Sensitive and Selective Fluorescent “Turn-On” Probe for Endogenous Detection of HOCl and Reactive Halogenated Species

**DOI:** 10.3390/antiox11091719

**Published:** 2022-08-30

**Authors:** Veronika E. Reut, Stanislav O. Kozlov, Igor V. Kudryavtsev, Natalya A. Grudinina, Valeria A. Kostevich, Nikolay P. Gorbunov, Daria V. Grigorieva, Julia A. Kalvinkovskaya, Sergey B. Bushuk, Elena Yu Varfolomeeva, Natalia D. Fedorova, Irina V. Gorudko, Oleg M. Panasenko, Vadim B. Vasilyev, Alexey V. Sokolov

**Affiliations:** 1Department of Biophysics, Belarusian State University, 220030 Minsk, Belarus; 2Department of Molecular Genetics, Institute of Experimental Medicine, 197376 Saint-Petersburg, Russia; 3Department of Immunology, Institute of Experimental Medicine, 197376 Saint-Petersburg, Russia; 4Department of Biophysics, Federal Research and Clinical Center of Physical-Chemical Medicine of Federal Medical Biological Agency, 119435 Moscow, Russia; 5Stepanov Institute of Physics, National Academy of Sciences of Belarus, 220072 Minsk, Belarus; 6SSPA “Optics, Optoelectronics, and Laser Technology”, 220072 Minsk, Belarus; 7Petersburg Nuclear Physics Institute named by B.P. Konstantinov of National Research Centre “Kurchatov Institute”, 188300 Gatchina, Russia

**Keywords:** hypochlorous acid, myeloperoxidase, neutrophils, fluorescent probes, reactive halogen species, phenoxazine

## Abstract

Hypochlorous acid (HOCl) derived from hydrogen peroxide and chloride anion by myeloperoxidase (MPO) plays a significant role in physiological and pathological processes. Herein we report a phenoxazine-based fluorescent probe Celestine Blue B (CB) that is applicable for HOCl detection in living cells and for assaying the chlorinating activity of MPO. A remarkable selectivity and sensitivity (limit of detection is 32 nM), along with a rapid “turn-on” response of CB to HOCl was demonstrated. Furthermore, the probe was able to detect endogenous HOCl and reactive halogenated species by fluorescence spectroscopy, confocal microscopy, and flow cytometry techniques. Hence, CB is a promising tool for investigating the role of HOCl in health and disease and for screening the drugs capable of regulating MPO activity.

## 1. Introduction

Neutrophils are the most abundant fraction of the leukocytes (50–70% of total circulating leukocytes in humans) [1]. They migrate to the site of inflammation and are able to perform a wide range of antimicrobial activities. Their functional responses are mostly associated with the initiation of respiratory burst and release of the content of their numerous granules. Neutrophils are professional phagocytes, however, the phagocytic process in neutrophils differs from that in macrophages [2]. The key role in neutrophil antimicrobial functions belongs to myeloperoxidase (MPO), an enzyme of azurophilic granules. MPO release can occur due to phagolysosomal leakage, cell lysis, excessive secretion or neutrophil extracellular traps (NETs) formation. MPO associated with NETs is active and mediates bacterial killing in the presence of hydrogen peroxide (H_2_O_2_) due to mainly hypochlorous acid (HOCl) formation [3,4]. HOCl underlies the antibacterial activity of MPO and provides an effective elimination of pathogens. Hypobromous and hypoiodous acids, along with hypothiocyanous acid, may also be produced in a halogenation cycle from the respective precursors. However, at physiological chloride concentrations, the main product of MPO will be HOCl. Otherwise, accomplishing the peroxidase cycle, MPO catalyzes the successive one-electron oxidation of substrates, e.g., tyrosine, urate, nitrite, ascorbate, etc. [4,5].

At the same time, increased concentrations of MPO as well as of chlorination biomarkers (3-chloro-tyrosin; 5-chloro-uracil), were reported in the development of oxidative/halogenative stress. Such a condition accompanies the inflammatory process in case of myocardial infarction, rheumatoid arthritis, renal insufficiency, cystic fibrosis, etc. [6,7,8,9]. Therefore, available and sensitive methods which allow detecting HOCl synthesis both inside single cells and in the extracellular space are needed.

Since neutrophil activation leads to the production of various reactive oxygen (ROS), halogen (RHS), nitrogen (RNS) species and other molecules, it is important to distinguish the products of MPO catalysis among other oxidants. By this point, different methods of neutrophil MPO activity analysis exist. Currently, there are two main approaches. The first one is the analysis of MPO activity using mass spectrometry [10,11,12,13], based on the detection of chlorinated amino acids, DNA/RNA bases or lipids. The second one relies on the usage of fluorescent and/or colorimetric probes specific to HOCl. It allows for a continuous analysis of MPO activity in living neutrophils and the detection of the main sites of HOCl production in cell suspensions, or even in tissues, using fluorescent probes. Various techniques aimed at the detection of MPO activity in vivo were also suggested, including usage of near-infrared probes [14], two-photon probes [15], quenched nanoparticles [16], and magnetic resonance imaging [17]. Some more details regarding these methodological approaches are reviewed in Huang et al. [18]. Nevertheless, the development of new ways of MPO activity detection in vitro also remains important.

Nowadays the problem of searching for a specific and sensitive probe for HOCl is being actively studied. Some of the commonly used probes, such as 2-nitro-5-sulfidobenzoate (TNB) [19,20], hydroethidine [21], monochlorodimedon [22] and aminophenyl fluorescein (APF) [23] could also be oxidized by other oxidants or be used as a substrate for the MPO peroxidase cycle. Moreover, selectivity of the majority of existing probes may be rather questionable due to a limited range of oxidants used in selective tests [12]. It is worth noting that neutrophils contain a large amount (20–50 mM) of taurine (Tau) [24,25,26], which is converted into Tau-N-chloramine [27] upon reaction with HOCl (rate constant ~3 × 10^5^ M^−1^ s^−1^ [28]). It still has oxidative potential, yet a number of fluorescence probes that react with HOCl do not react with Tau-N-chloramine. This phenomenon restricts the usage of such probes in evaluation of HOCl production by activated neutrophils.

Some currently available probes are aimed at «naked-eye» detection [29,30] or are based on fluorescence-quenching [31,32,33], limiting their application in practice of HOCl measurements in single cells using microscopy or flow cytometry. Finally, even though there are novel fluorescent probes with high selectivity and sensitivity declared, they are currently difficult approaches for the majority of research laboratories. Some fluorescent HOCl probes developed since 2016 and their characteristics are presented in Appendix A. Some more information could be found in numerous review articles [34,35,36,37].

Previously we proposed a spectrophotometric method of measuring the kinetics of HOCl production by monitoring the oxidation of the dye Celestine blue B—C.I.51050 (CB)—in the presence of Tau [38]. We proved that CB selectively reacts with HOCl and Tau-N-chloramine and becomes oxidized to glycol, which makes it change color to pink. Neither superoxide radical anion (O_2_^−^) nor H_2_O_2_ oxidize CB. It was also shown that the oxidation of CB by HOCl and Tau-N-chloramine occurs with high-rate constants, such as 2.5 × 10^4^ and 1.2 × 10^5^ M^−1^ s^−1^, respectively.

In this study we made use of CB as a fluorescent probe of a “turn-on” type for HOCl detection in cellular systems by confocal microscopy, flow cytometry, and fluorescence spectroscopy assays.

## 2. Materials and Methods

### 2.1. Chemicals

The following chemicals were used: CB, APF, 4’,6-diamidine-2-phenyl indole (DAPI), 4-aminobenzoic acid hydrazide (ABAH), glycerol, sodium hypochlorite (NaOCl) in solution (1.78 M), KO_2_, xanthine oxidase (XO), xanthine (X), superoxide dismutase (SOD), mannitol, poly-L-lysine in solution (0.1%), Tau, phorbol 12-myristate 13-acetate (PMA), EDTA, scopoletin, type II horseradish peroxidase (HRP), N-formyl-methionyl-leucyl-phenylalanine (fMLP), cytochalasin *b* (cyt*b*), Hoechst 33342, diphenyleneiodonium chloride (DPI), histopaque-1077 and sodium azide (NaN_3_) (all from Sigma-Aldrich, St. Louis, MO, USA). Other reagents were from Reakhim (Russia) and Belmedpreparaty (Belarus). Ceruloplasmin (CP) was isolated from the trisodium citrate-stabilized blood plasma of healthy donors as described at [39]. MPO was isolated from human neutrophil extract as described at [39]. Plant lectins from *Canavalia ensiformis* (Con A), *Caragana arborescens* (CAA), *Glycine hispida* (SBA), *Phaseolus vulgaris* (PHA-L), *Sambucus nigra* (SNA), and *Triticum vulgaris* (WGA) were from Lektinotest (Lviv, Ukraine). Solutions were prepared using apyrogenic deionized water with a specific resistance 18.2 MΩ × cm (Mediana-Filter, Russia). The pH of buffer solutions was measured using a portative pHmeter (Aquilon pH-410, Russia) having the precision of measurements of 0.01 pH.

A concentrated solution of CB (1 mM) was prepared in 20% aqueous glycerol (*v*/*v*) and stored in the dark at +4 °C.

The phosphate-buffered saline (PBS) containing 10 mM Na_2_HPO_4_/KH_2_PO_4_, 137 mM NaCl, 2.7 mM KCl, and with a pH of 7.4 served as a buffer.

### 2.2. Preparation of Human Neutrophils

Venous blood samples were obtained from healthy donors at the Republican Scientific and Practical Centre for Transfusiology and Medical Biotechnologies (Minsk, Belarus). Then, 3.8% (*w*/*v*) trisodium citrate was used as an anticoagulant at a ratio of 9:1. Neutrophils were isolated by centrifugation in the histopaque-1077 density gradient as described at [40]. Cells were suspended in PBS (pH 7.4), containing 5 mM D-glucose, and stored at +4 °C. The percentage of neutrophils in the cell preparations was >95%, and the viability was >95%, determined by trypan blue assay. All isolation procedures were performed at room temperature.

### 2.3. Spectrofluorimetry

Fluorescence intensity was measured using a flatbed fluorimeter CLARIOStar (BMG LABTECH, Germany) or CM 2203 spectrofluorimeter (SOLAR, Belarus). CB (20 μM) and HOCl (0.5–50 μM) in PBS were mixed. Excitation and emission maxima of CB and of its HOCl-oxidized form were scanned sequentially (220–520 nm for excitation and 550–610 nm for emission). The maxima chosen (excitation 430 nm, emission 590 nm) were used to determine fluorescence intensity depending on the HOCl concentration.

### 2.4. Preparation of ROS, RHS, RNS

O_2_^−^ was generated by two ways: by introducing a weighed portion of KO_2_ into the test solution to reach the final concentration of 100 μM, and also in the XO/X system containing 140 mg/L XO and 1 mM X as described in [41].

Hydroxyl radical (•OH) was formed in the Fenton reaction by mixing 1 mM FeSO_4_ × 7H_2_O with 100 μM H_2_O_2_ as described in [23]. Stock solutions of HOCl and H_2_O_2_were prepared by diluting a commercial solution of NaOCl and H_2_O_2_ (30%) in deionized water. Solution of nitrite (NO_2_^−^) was obtained from NaNO_2_, dissolved in deionized water. Peroxynitrite (ONOO^−^) was synthesised from H_2_O_2_ and NaNO_2_, residual H_2_O_2_ was removed by MnO_2_. The concentration of ONOO^−^ was determined spectrophotometrically at 302 nm, **ε**_302_ = 1670 M^−1^ cm^−1^ [42].

### 2.5. Assessing the Limit of Detection (LOD)

The fluorescence spectra of native probes (20 μM) were measured eleven times, and the standard deviation (σ) of a blank measurement was deduced. Next, different equivalents of HOCl were added and the fluorescence intensity at 590 nm (excitation at 430 nm) was recordedimmediately. Then, the data obtained were plotted as the increasing concentrations of HOCl, and the slope (k) was obtained. The LOD was calculated using the following equation: LOD = 3σ/k [43].

### 2.6. Chlorinating Activity of MPO in Solution and Specific Immuno-Extraction Followed by Enzymatic Detection (SIEFED) Assay

Chlorinating activity of MPO was assayed in 96-well plate: to 100 µL serial 2-fold dilutions of MPO from 125 ng/mL 100 mM Na–acetate buffer, with a pH 5.5, 100 µL of 2× stock developer containing 400 µM CB, 100 µM H_2_O_2_, 4 mM Tau, 10 µM KI, 500 mM NaCl, and 100 mM Na–acetate buffer, at pH 5.5, were added. To assess the generation of HOCl by MPO a 250 µM solution of HOCl in 100 mM Na–acetate buffer, pH 5.5, was diluted by half in series with 100 µL of H_2_O_2_-free 2× stock developer. The plate was incubated for 20 min at 37 °C with continuous stirring at 300 rpm. The fluorescence intensity of CB-oxidized product was measured at 573–583 nm (excitation at 482–492 nm) in the multimodal plate reader CLARIOstar (BMG LABTECH, Germany) using the «Fluorescence intensity» mode.

To compare MPO concentration estimated by enzyme-linked immunosorbent assay (ELISA) [44] and by the CB-based assay of MPO activity by SIEFED, samples of the whole blood were incubated either with PMA or without the activator. A blood sample (10 mL) stabilized by heparin was divided in two aliquots (one of which was supplemented with 50 nM PMA) and incubated for 4 h at 37 °C with continuous stirring at 300 rpm. Every 30 min 300 µL of blood was sampled and centrifuged 5 min at 1500× *g* to separate plasma for the MPO assay. For the SIEFED assay, MPO was adsorbed using monoclonal antibodies against MPO (clone 2F7) obtained by immunizing mice and producing a hybridoma as described earlier [44]. Then, 100 µL of 2F7 antibodies (5 mg/L) in borate-buffered saline (BBS) was dropped into the wells of standard polystyrene plates and incubated 2 h at 37 °C or overnight at 4 °C, after which three washings were done with BBS containing 0.05% Tween-20. Then, 100 µL of the blocking buffer (3% milk in PBS, 0.05% Tween-20) was added into the wells. After 1 h of incubation, plasma or MPO samples (10 µL) were added into each well. The initial concentration of the standard MPO solution was 2000 ng/mL, the minimum was 31.5 ng/mL. The plates again were incubated for 1 h at 37 °C with continuous stirring at 300 rpm, after which three washings followed with BBS containing 0.05% Tween-20. Then, 100 µL of the developer containing 200 µM CB, 50µMH_2_O_2_, 2 mM Tau, 5 µM KI, 250 mM NaCl, 100 mM Na–acetate buffer, pH 5.5, were added. The plate again was incubated for 60 min at 37 °C with continuous stirring at 300 rpm. The fluorescence intensity of CB-oxidized product was measured. Adding 250 mM NaCl and 100 mM Na–acetate buffer to the developer solution allowed us to neutralize the inhibitory effect of ceruloplasmin (CP) in plasma [39].

### 2.7. H_2_O_2_ Production by Neutrophils Measured Using Scopoletin

H_2_O_2_ generation by neutrophils was measured using the scopoletin/HRP fluorescent technique [45]. Briefly, the suspension of neutrophils (2 × 10^6^ cells/mL in PBS, containing 1 mM CaCl_2_, 0.5 mM MgCl_2_) was supplemented with 1 μM scopoletin (a fluorescent substrate of peroxidase), 20 μg/mL HRP, and 1 mM NaN_3_ (catalase and MPO inhibitor) and fMLP at various concentrations was added to activate plasma membrane NADPH oxidase. The H_2_O_2_-mediated oxidation of scopoletin was recorded as a decrease of its fluorescence at 460 nm (excitation was at 350 nm) during 20 min at 37 °C using a CM2203 spectrofluorimeter (SOLAR, Belarus).

### 2.8. Registration of HOCl Production by Activated Neutrophils

Neutrophils (10^5^ cells/mL) were incubated at 37 °C in a 96-well plate under continuous stirring (290 rpm; Biosan, Latvia) or 10^6^ cells/mL in a quartz cuvette of CM 2203 spectrofluorimeter (SOLAR, Belarus) in the presence of 20 μM CB and without it in the PBS, containing 1 mM CaCl_2_, 0.5 mM MgCl_2_, 5 mM D-glucose, and 20 mM Tau. After a short pre-incubation, cells were activated with PMA (0–100 nM), fMLP (0–2 μM), WGA (50 mg/L), CAA (75 mg/L), PHA-L (100 mg/L), Con A (100 mg/L), SNA (75 mg/L), or SBA (100 mg/L). Activation of neutrophils with fMLP and lectins was preceded by pre-incubation of cells with cyt *b* (2.5 mg/L) to provide the degranulation of the azurophilic granules. Fluorescent response via fMLP/lectins + cyt *b* system was compared with the appearance of fMLP/lectins in the absence of cyt *b*. To minimize the neutrophil proteins’ contribution in the fluorescence assay, the excitation was performed at 460 nm.

HOCl generation by neutrophils was also registered using APF. Neutrophils were incubated with PMA (50 nM) in the presence of 2 μM of APF with continuous recording of the fluorescence intensity changes (excitation at 490 nm, emission at 520 nm) [23,46].

In both cases after 20 min of neutrophils stimulation at 37 °C, the fluorescence intensity was compared in the presence and absence of ROS/RHS production inhibitors or their scavengers.

### 2.9. Confocal Microscopy Assay of Fixed Cells

Neutrophils for confocal microscopy were prepared as follows. Coverslips (18 × 18 mm) were incubated for 1 h in 0.1% poly-L-lysine solution, rinsed with water and dried at room temperature. A suspension of neutrophils (1.25 × 10^5^ cells/mL) with various combinations of CB (20 μM) or APF (5 μM) and PMA (50 nM) was applied on the treated coverslips and incubated for 30 min at 37 °C. The glasses were carefully washed with PBS, after which cells were fixed in 4% paraformaldehyde (*w*/*v*) within 10 min at room temperature. In the following 5 min preparations were stained with DAPI (0.36 μM) at room temperature in the dark, after which the coverslips were washed two times with PBS and once with deionized water and mounted on the slides using Fluoromount^TM^ aqueous mounting medium. Images were obtained with an inverted confocal laser-scanning microscope, LSM 510 META or LSM 510 NLO (Carl Zeiss, Germany), supplied with an objective-lens Plan-Apochromat 20×/0.8 M27 (LSM 510 META) or Zeiss Plan-Neofluar 40×/0.75 (LSM 510 NLO, Carl Zeiss, Germany). Both CB and APF were excited at 488 nm, and emission was registered using a filter cutting off the waves below 505 nm. DAPI was excited at 405 nm (LSM 510 META) or DAPI was two-photon excited at 810 nm (effective excitation at 405 nm) by femtosecond laser radiation (Spectra-Physics Tsunami, pulse duration 100 fs, repetition rate 82 MHz, LSM 510 NLO) and its fluorescence was registered at 420–480 nm. Images were elaborated by the software provided with the microscope.

### 2.10. Confocal Microscopy Assay of Viable Cells

Neutrophils for confocal microscopy were prepared as follows. An amount of 0.1% poly-L-lysine solution was kept for 1 h in optical Petri dishes (d = 35 mm), after which they were rinsed with water and dried at room temperature. A suspension of isolated neutrophils was applied on the treated optical Petri dishes and incubated for 30 min at 37 °C. After that samples were treated with 20 μM CB and 20 mM Tau. To view the precise localization of RHS in neutrophils, we combined CB and Hoechst 33,342 for staining the RHS and nucleus, respectively. Hoechst 33,342 was added before the 30-min incubation mentioned above (for better staining). To obtain the maximum RHS output we added PMA at a concentration of 16.2 nM. Images were obtained with an inverted confocal laser-scanning microscope TCS SP5 SMD FLCS (Leica, Germany). CB was excited at 488 nm, and the emission was registered using a filter cutting off the waves below 515 nm. Hoechst 33,342 was excited at 405 nm and its fluorescence was registered in the range of 425–475 nm (band pass filter 450 ± 25 nm). The scanning started 15 min after PMA was added and lasted for 1 h. Images were elaborated by the software provided with the microscope.

### 2.11. Flow Cytometry Assay

The fluorescence intensity of CB in neutrophils (excitation at 488 nm, emission at 620 ± 30 nm, 22 MW) was measured using a flow cytometer, Navios^TM^ (Beckman Coulter Inc., Brea, CA, USA). Neutrophils (10^6^ cells/mL) were incubated for 30 min with different combinations of CB (20 μM), PMA (50 nM), and ABAH (50 μM), washed by centrifugation (400× *g*, 7 min), and the samples were analyzed by flow cytometry. Each sample contained at least 10,000 neutrophils, and the data obtained were elaborated using Kaluza Flow 2.3 (Beckman Coulter Inc., Brea, CA, USA) software.

### 2.12. Statistical Analysis

Statistical and graphical processing of data was performed using the software package OriginPro 2016 (OriginLab Corporation, Northampton, MA, USA). The experiments were repeated thrice (n = 3, if not specified otherwise). The results are presented as mean ± SD, if not stated otherwise. Significant differences between means were identified using Student’s *t*-test or one-way ANOVA followed by the posthoc Tukey’s HSD test to analyze differences between means. Values of *p* < 0.05 were considered to indicate statistically significant differences. To assess the compliance of actual and calculation data with linear dependence, the coefficient of determination R^2^ (normalized in the range from 0 to 1) was computed.

## 3. Results

### 3.1. Fluorescence Response of Celestine Blue B to HOCl

Measurements of the fluorescence intensity provided by oxidized CB at varying HOCl concentrations in PBS (pH 7.4) and in its absence allowed assessing the ability of CB to recognize HOCl. Upon CB treatment with 50 μM HOCl, numerous excitation peaks were found at 260, 290, 312, 365 and 429 nm (Figure 1A). The fluorescence spectra of CB in the presence of varied concentrations of HOCl are shown in Figure 1B. There was no fluorescence signal detected in excitation and emission spectra of the non-oxidized CB in the absence of HOCl (Figure 1B, blue line). At HOCl concentrations below 4 µM, no oxidation of the probe was observed. The relation between the fluorescence intensity of oxidized CB (20 μM; excitation at 430 nm, emission at 590 nm) and HOCl concentration (4–50 μM, R^2^ = 0.999) was linear (Figure 1C). The LOD calculated for HOCl was 32 nM and was similar to that determined for the majority of other existing probes (see Appendix A). The time–resolution curve plotted for the CB reaction with 1 equivalent of HOCl indicates a fast response of the probe (reaction was complete within 30 s, Appendix A). The selectivity of CB towards HOCl in comparison with other ROS/RNS was examined. As shown in Figure 1D, 2 equivalents of HOCl gave a ~45-fold enhancement in fluorescence intensity. None of other analytes (H_2_O_2_, O_2_^−^, •OH, NO_2_^−^, ONOO^−^) caused such obvious changes in the fluorescence intensity of CB (Figure 1D). The above results evidenced that CB could specifically distinguish HOCl instead of the other ROS/RNS.

Assaying MPO chlorinating activity in the developer solution (see Section 2.6) and comparing the fluorescence intensity in serial dilutions of HOCl allows one to measure the rate of HOCl production by MPO (Figure 2A). The SIEFED-based assay allows us to estimate the active MPO in 10 μL of samples containing 31–2000 ng/mL enzyme (Figure 2B).

The results of active MPO detection accomplished 30 min after separating plasma from heparin-stabilized blood incubated with PMA and without the activator at 37 °C for 4 h are demonstrated in Figure 2C.

### 3.2. Application of CB for Endogenous HOCl Detection in Living Cells

#### 3.2.1. Fluorescence Spectroscopy

Neutrophil activation can be triggered by a multitude of stimuli that activate various cellular signaling pathways, which allows one to analyze the mechanisms of neutrophil activation. Several agents, including fMLP and PMA, are commonly used to study neutrophil activation in vitro. PMA, a structural analogue of diacylglycerol, is a non-physiological stimulus. It is able to enter cells and directly activate protein kinase C, followed by a prolonged extra- and intracellular ROS generation [47,48]. In our experiments incubation of neutrophils in the medium containing PMA and CB led to a “turned on” fluorescent signal from oxidized CB, which was both time- and PMA concentration-dependent (Figure 3A,B). A lag-phase for the onset of fluorescence is due to two processes that need to happen: assembly and activation of NADPH-oxidase, which is fast [49], and MPO degranulation, which is a slow process [50,51]. In the absence of PMA, the fluorescence intensity of oxidized CB in the neutrophil suspension remained close to the background level (Figure 3A, curve 1). Pre-treatment of neutrophils with ABAH, a well-known MPO inhibitor [52], also brought the fluorescence intensity to the background value (Figure 3B).

Other inhibitors of ROS production were tested, such as SOD, DPI, mannitol and CP. All of them except mannitol suppressed CB oxidation by PMA-activated neutrophils (Figure 3C). The results obtained are in agreement with those for APF; however, it is worth noting that Tau suppressed APF fluorescence (Figure 3D).

fMLP is another neutrophil activator that engages the surface formyl peptide receptor (a G-protein-coupled receptor), which results in a short-term O_2_•^−^ production by NADPH-oxidase, but not in azurophilic granules’ egress and degranulation [53,54]. To induce exocytosis of MPO by fMLP in vitro, it is necessary to pre-incubate neutrophils with cyt *b*, which destroys cortical filamentous actin (F-actin) [54,55].

When neutrophils were activated by fMLP with cyt *b* pre-incubation, the fluorescence of oxidized CB became more intense (Figure 4A, curves 2–5). In the absence of cyt *b*, this effect was not observed (Figure 4A, curve 1). However, in experiments with scopoletin as a substrate for HRP it was shown that fMLP caused the dose-dependent production of H_2_O_2_ by activated neutrophils (Figure 4B).

Then, we studied the fluorescence intensity of CB in a suspension of neutrophils activated by lectins, which, as we have shown earlier using the scopoletin test, induce the assembly of NADPH oxidase [45]. In the present work we used CB to investigate the influence of WGA, CAA, Con A, SNA, SBA, and PHA-L on HOCl production by neutrophils. The CB fluorescent response via PMA and fMLP/lectins + cyt *b* neutrophil activation is provided in the Table 1.

The most active stimulants of HOCl production by neutrophils among the used lectins were WGA and CAA, whereas SNA and SBA were unable to stimulate HOCl production. It should be noted that, as in the case with fMLP, the fluorescence intensity of CB in the suspensions of neutrophils, activated by lectins, increased only when cells had been pre-incubated with cyt *b* (2.5 mg/L). Therefore, CB can be used to detect the production of HOCl by neutrophils that is caused by various stimuli.

#### 3.2.2. Confocal Microscopy

Next, the confocal imaging was applied to assess the applicability of CB for endogenous HOCl detection in human polymorphonuclear neutrophils. The neutrophils incubated with CB revealed a very low basal level of fluorescence, most probably arising from their spontaneous activation upon adhesion to the glass (Figure 5a). Upon incubation of neutrophils with PMA, a significant fluorescence signal enhancement of oxidized CB was obtained. A video animation of neutrophil activation without and with Hoechst 33342-stained nuclei is presented as Appendix A.

It is clearly known that the prolonged activation of neutrophils with PMA leads to NET formation. Such structures are currently viewed not only as part of anti-pathogen defenses [56], but also as factors of autoinflammatory and even autoimmune diseases [57]. To induce NET formation, neutrophils were stimulated with PMA for 30 min at 37 °C. Local sites with a high fluorescence intensity of oxidized CB were clearly seen (Figure 5b). These sites were co-localized with DNA stretches protruding from the cells, corresponding to ROS-dependent NETosis by the cells. A GIF animation of the layer-by-layer visualization of images obtained by confocal microscopy is also presented in Appendix A.

#### 3.2.3. Flow Cytometry

Finally, we decided to explore the applicability of CB in flow cytometry experiments. Neutrophils were pre-incubated with 20 μM of CB, then PMA and ABAH were added in various combinations, which was followed by flow cytometry analysis. A seven-fold increase of the fluorescence intensity was observed (*p* < 0.05) when cells were incubated with CB (Figure 6A). When cells were pre-treated with PMA (50 nM), a strong fluorescence “turn-on” signal appeared (Figure 6B). Adding ABAH led to a 1.5-fold decrease in the fluorescence intensity in both the cases of the unstimulated and stimulated neutrophils (Figure 6C,D).

## 4. Discussion and Conclusions

CB has been used in histology assays as a cationic dye for nuclei and tissue staining as an alternative for hematoxylin and eosin procedures [58,59,60]. Recently our group demonstrated that CB can be applied to detect HOCl and Tau-N-chloramine produced by isolated MPO and in cell suspension, using the absorption method [38].

Our results show that CB oxidation by HOCl leads to a fluorescent “turn-on” response, which makes CB a useful tool for the rapid HOCl detection by fluorescence technique. The probe exhibited an excellent Stokes shift, excitation in blue and emission in orange channels, and good sensitivity (LOD = 32 nM) and selectivity for HOCl and reactive halogenated species over other ROS/RNS (Figure 1). These results are comparable with properties of the majority of existing probes for HOCl (Appendix A). Moreover, CB is largely available as a commercial product.

The experiments accomplished showed that the SEIFED method allows using CB for the estimation of MPO concentration in the blood plasma samples activated by PMA. A significant correlation between SEIFED and ELISA has been shown (Figure 2).

We also explored CB’s applicability for the detection of endogenous HOCl and reactive halogenated species in a suspension of living cells using receptor-dependent (fMLP, lectins) [45,55] and receptor-independent (PMA) mechanisms of neutrophil NADPH oxidase activation [47]. When cells were stimulated with PMA, a dose-dependent fluorescent “turn-on” signal of oxidized CB was observed (Figure 3A). In the presence of ABAH, a well-known MPO inhibitor, the fluorescence intensity of CB in a suspension of PMA-stimulated neutrophils was strongly reduced (Figure 3B), indicating the ability of CB to measure HOCl and reactive halogenated species in a suspension of activated neutrophils.

The results obtained for the receptor-dependent mechanism of neutrophil activation have shown that both H_2_O_2_ production and exocytosis of MPO are prerequisites for an increase in the fluorescence intensity of the probe, which is indicative of HOCl and reactive halogenated species’ production in the cell suspension (Figure 4, Table 1).It was shown that fMLP and several plant lectins cause an increase of the CB fluorescence in neutrophil suspension only upon pre-incubation of the cells with cyt *b*. This occurs, most probably, due to the release of MPO from azurophilic granules and, consequently, the production of HOCl and reactive halogenated species accompanied by blocking the microfilament system of cells (Table 1) [53,61].

Hence, CB can be applied for discrimination between ROS production (as a result of NADPH-oxidase activation) and RHS production (as a result of both NADPH-oxidase activation and MPO exocytosis) by activated neutrophils with fMLP and lectins in the absence and the presence of cyt *b*, respectively.

Furthermore, CB allowed viewing the formation of NETs using confocal microscopy. Pre-treatment of neutrophils with PMA led to the release of NETs, marked by an increase of the local CB fluorescence both in separately viewed activated cells and at the sites where intense CB fluorescence co-localized with NETs (Figure 5). It is known that under certain conditions, activated neutrophils release NETs along with the granules’ content [56]. As a result, numerous antimicrobial peptides and proteins, including HOCl-producing MPO [56,62], have become associated with NETs. Therefore, CB allows us to detect sites of HOCl and reactive halogenated species production in activated neutrophils. Moreover, CB fluorescence intensity increment was registered in the flow cytometry assay (Figure 6). These results proved that CB is able to cross neutrophilic plasma membrane. Thuswise, CB allows to observe HOCl and reactive halogenated species production in single cells and to investigate the influence of MPO inhibitors on this process.

In summation, a novel fluorescent probe CB for selective and specific HOCl and RHS detection is proposed. It is important to note that CB should applied only for in vitro studies, with MPO-containing cells, and control experiments with MPO inhibitors (such as ABAH) need to be used to confirm that any changes of fluorescence are due to MPO-derived oxidants (RHS). In our previous papers we showed the resistance of CB to HOSCN oxidation and a practically equal HOCl reactivity with HOBr [38], but the Km ofNaBr for MPO at pH 7.4 was estimated as being only 2.1 mM [63]. CB has successfully passed tests for endogenous HOCl imaging with fluorescence spectroscopy, confocal microscopy and flow cytometry techniques. Among the advantages of CB are its pronounced stability, specificity to HOCl and Tau-N-chloramine, and also its high availability of the dye in comparison with other fluorescent probes described in the literature [37]. Our results showed that CB can be applied as a reliable “turn-on” fluorescent probe for endogenous HOCl and reactive halogenated species’ detection in living neutrophils activated by a broad range of agonists, as well as for investigating the impact of various drugs on this process in physiological and pathological conditions.

## Figures and Tables

**Figure 1 antioxidants-11-01719-f001:**
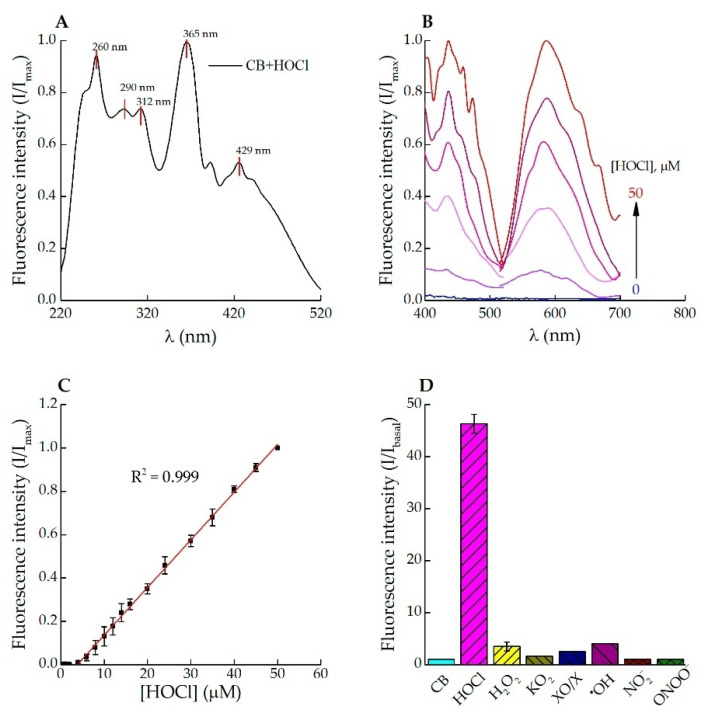
Fluorescence spectra of CB (20 μM) in the presence of HOCl. (**A**) Full normalized excitation spectra of CB after reaction with 20 μM HOCl (λem = 590 nm); (**B**) Excitation spectra at 400–520 nm (λem = 590 nm) and emission spectra at 515–700 nm (λex = 430 nm) of CB in the absence (blue line) and in the presence of 10–50 μM HOCl (from purple to red lines); (**C**) Fluorescence intensity of CB as the dependence on HOCl concentration in the range of 1 to 50 μM (λex = 430 nm, λem = 590 nm); (**D**) Fluorescence responses of CB to HOCl (50 μM NaOCl), H_2_O_2_ (100 μM), KO_2_ (100 μM), XO/X (140 mg/L XO and 1 mM X), •OH (1 mM FeSO_4_ × 7 H_2_O and 100 μM H_2_O_2_), NO_2_^−^ (100 μM NaNO_2_), ONOO^−^ (100 μM). All reactions carried out in PBS (pH 7.4), T = 25 °C, λex = 430 nm, λem = 590 nm.

**Figure 2 antioxidants-11-01719-f002:**
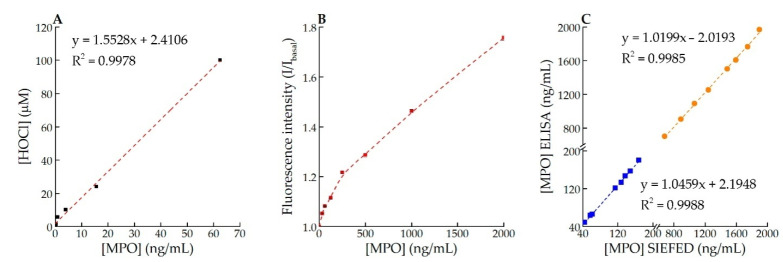
Assessing the MPO chlorinating activity in solution and in the SIEFED assay: (**A**) Production of HOCl (μM) during 20 min incubation of MPO (0–62.5 ng/mL) with the developer containing 200 µM CB, 50 µM H_2_O_2_, 2 mM Tau, 5 µM KI, 250 mM NaCl, 100 mM Na–acetate buffer, pH 5.5; (**B**) I/I_basal_ (λex = 487 nm, λem = 578 nm) obtained for MPO calibrators (31.3–2000 ng/mL) in the SIEFED assay; (**C**) Results of MPO assay (by SIEFED—abscissa and by ELISA—ordinate) in plasma samples taken every 30 min from heparin-stabilized blood incubated at 37 °C during 4 h with 50 nM PMA (orange points) and without activator (blue points).

**Figure 3 antioxidants-11-01719-f003:**
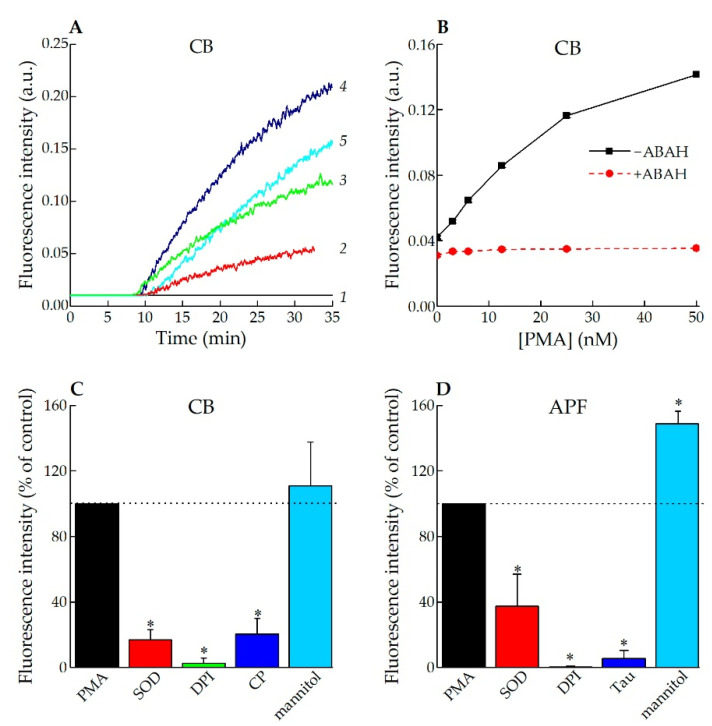
PMA-induced neutrophil activation. (**A**) Time-dependent changes in the fluorescence intensity of CB upon neutrophil activation. Neutrophils (10^6^ cells/mL) were pre-incubated with different concentrations of CB: 5 µM (2), 10 µM (3), 20 µM (4), 40 µM (5) in the absence (1) and presence (2–5) of 50 nM PMA. (PBS + 20 mM Tau, pH 7.4, T = 37 °C; λex = 460 nm, λem = 590 nm); (**B**) CB (20 μM) fluorescence intensity after 60 min of neutrophils activation with PMA (0–50 nM) in the absence (solid line) and presence of an MPO inhibitor (50 μM ABAH; dashed line). (PBS + 20 mM Tau, pH 7.4, T = 37 °C; λex = 460 nm, λem = 590 nm); (**C**) CB (20 μM) fluorescence intensity in PMA-activated neutrophil suspension in the absence and presence of ROS-generating scavengers and inhibitors. (PBS + 20 mM Tau, pH 7.4, T = 37 °C; λex = 460 nm, λem = 590 nm); * *p* < 0.05 versus the effect of PMA; (**D**) APF (2 μM) fluorescence intensity in PMA-activated neutrophil suspension in the absence and presence of ROS-generating scavengers and inhibitors. (PBS, pH 7.4, T = 37 °C; λex = 490 nm, λem = 525 nm); * *p* < 0.05 versus the effect of PMA.

**Figure 4 antioxidants-11-01719-f004:**
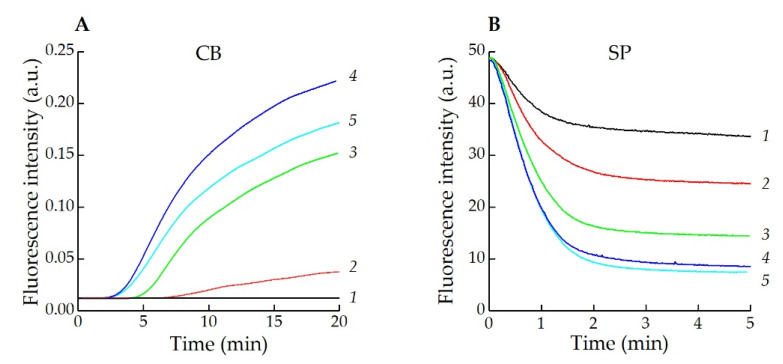
fMLP-induced neutrophil activation. (**A**) Kinetic curves of CB fluorescence reflect the intensity increase. Neutrophils (10^6^ cells/mL) were pre-incubated with CB (20 μM) in the absence (1) and presence (2–5) of cyt *b* (2.5 mg/L) and different fMLP concentrations were added: 1 µM (1, 4), 0.25 µM (2), 0.5 µM (3), 2 µM (5) (PBS + 20 mM Tau, pH 7.4, T = 37 °C; λex = 460 nm, λem = 590 nm); (**B**) Typical kinetic curves of scopoletin oxidation. Neutrophils (2 × 10^6^ cells/mL) were pre-incubated with 1 μM scopoletin, 20 μg/mL HRP, 1 mM NaN_3_ and different fMLP concentrations were added: 0.5 µM (1), 1 µM (2), 2.5 µM (3), 5 µM (4), 10 µM (5). (PBS, pH 7.4, T = 37 °C; λex = 350 nm, λem = 460 nm).

**Figure 5 antioxidants-11-01719-f005:**
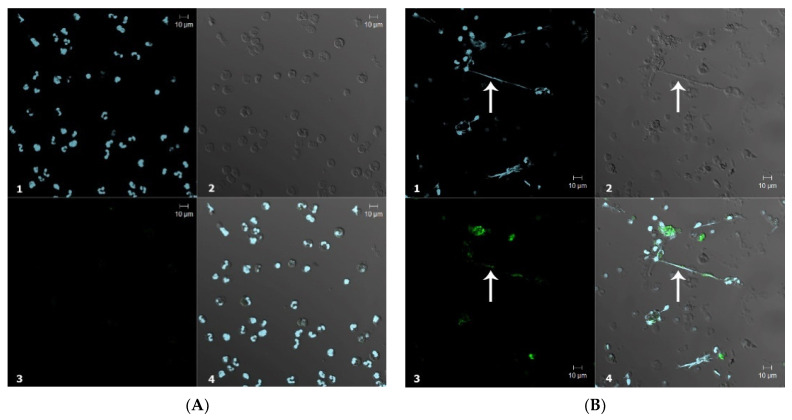
Confocal microscopy images of neutrophils incubated with 20 μM CB (**A**) in the absence of PMA; (**B**) in the presence of 50 nM PMA. White arrow indicates the NETs. 1—DAPI (blue), 2—differential interference contrast, 3—fluorescence of oxidized CB upon laser excitation at 488 nm (green), 4—merge of 1, 2 and 3.

**Figure 6 antioxidants-11-01719-f006:**
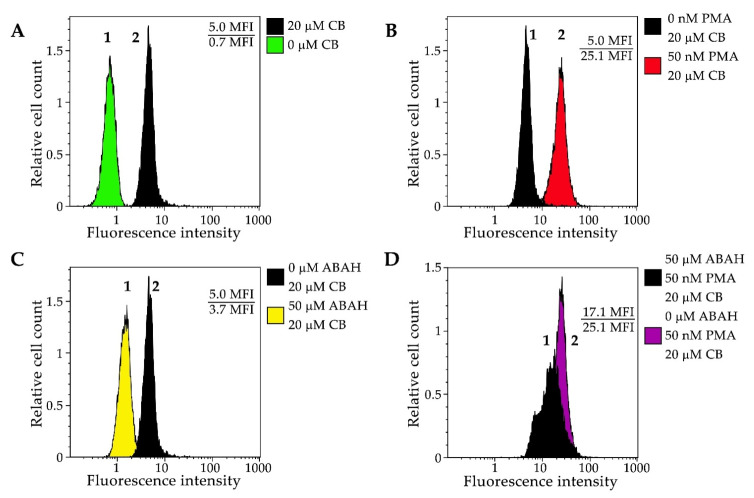
Histograms of CB fluorescence intensity in neutrophils (λem = 620 ± 30 nm) obtained by flow cytometry. (**A**) Cells in the absence of CB (1) or stained with 20 μM CB (2); (**B**) Cells in the absence of PMA (1), and those stimulated with 50 nM PMA (2) and stained with 20 μM CB; (**C**) and (**D**) Cells stained with 20 μM CB in the absence (2) and in the presence of 50 μM ABAH (1), before (**C**) and after stimulation with 50 nM PMA (**D**).

**Table 1 antioxidants-11-01719-t001:** Effect of various agonists on HOCl production by neutrophils, detected by CB.

Activation Stimuli	Concentration	Fluorescence Intensity (a.u.)
PMA ^1^	50 nM	0.139 ± 0.004 *
fMLP	1 μM	0.130 ± 0.028 *
WGA	50 mg/L	0.090 ± 0.003 *
CAA	75 mg/L	0.150 ± 0.045 *
PHA-L	100 mg/L	0.076 ± 0.020 *
Con A	100 mg/L	0.083 ± 0.044 *
SNA	75 mg/L	n/d ^2^
SBA	100 mg/L	n/d ^2^

^1^ Activation without cyt *b*. ^2^ Fluorescence signal was not detected. * *p* < 0.05 versus the basal level in the case of PMA or versus fMLP/lectins in the absence of cyt *b*. Neutrophils (10^6^ cells/mL) were pre-incubated with 20 μM CB and different stimuli were added. PBS + 20 mM Tau, pH 7.4, T = 37 °C; λex = 460 nm, λem = 590 nm.

## Data Availability

All of the data is contained within the article and the Appendix A.

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
