# Peer review of "New Application of the Commercially Available Dye Celestine Blue B as a Sensitive and Selective Fluorescent “Turn-On” Probe for Endogenous Detection of HOCl and Reactive Halogenated Species"

_antioxidants, 2022, doi:10.3390/antiox11091719_

Round 1
Reviewer 1 Report
The manuscript submitted by Sokolov et al. for publication in Antioxidants describes the development of a turn-on fluorescence assay for the detection of HOCl, using celestine 2 Blue B (CB) dye, which is selectively oxidized by the hypochloride leading to a fluorescent oxidation product. The usefulness of CB for endogenous HOCl imaging with fluorescence spectroscopy has been shown, using confocal microscopy and flow cytometry. CB appears to be more stable than other HOCl fluorescence probes described before and has a high specificity to HOCl and Tau-N-chloramine, a chlorination product formed by the action of HOCl on taurine.
The manuscript is well written, which a concise introduction describing the literature knowledge relevant to the subject; results are well described and presented, as well as, the discussion is performed adequately. Therefore, in my opinion the manuscript should be accepted for publication in Antioxidants; I have just some minor questions and suggestions as outlined below.
Line 75: Use the right IUPAC name for “5-thio-2-nitrobenzoic acid”. what you mean Line 89: What you want to say by "in that number" here?
Page 7, Figure 1C: The calibration curve does not pass to the origin. It appears that there is something like a lag-phase in [HOCl]. I believe that this should be discussed and explained.
Page 8, Figure 1A: This figure shows a lag-phase for the onset of fluorescence of almost 10 min. It would be interesting to mention this in the text and to discuss/explain it.
Page 9, Table 1: The experimental condition in which these assays have been preformed should be given here in the legend.
Author Response
We are grateful to both referees for a thorough study of our work and below we present answers to their questions.
Review â„–1
Line 75: Use the right IUPAC name for “5-thio-2-nitrobenzoic acid”. what you mean Line 89: What you want to say by "in that number" here?
Thank you for your comment. This oversight has been corrected in the text of the article.
"in that number" means "including", "for example". Edited for better reading.
Page 7, Figure 1C: The calibration curve does not pass to the origin. It appears that there is something like a lag-phase in [HOCl]. I believe that this should be discussed and explained.
The study of FI changes was performed immediately after HOCl addition. Under these conditions, at HOCl concentrations below 4 µM, no oxidation of the probe was observed. As described in «Materials and Methods» section, LOD was calculated on the linear plot of concentration curve (FI as HOCl function). In our case, it was 4-50 μM. We added this note at the text of the article (page 6).
Page 8, Figure 1A: This figure shows a lag-phase for the onset of fluorescence of almost 10 min. It would be interesting to mention this in the text and to discuss/explain it.
CB is oxidized in neutrophil suspension due to two processes: assembly and activation of NADPH oxidase [Caldefie-Chézet F, Walrand S, Moinard C, Tridon A, Chassagne J, Vasson MP. Is the neutrophil reactive oxygen species production measured by luminol and lucigenin chemiluminescence intra or extracellular? Comparison with DCFH-DA flow cytometry and cytochrome c reduction. Clin Chim Acta. 2002 May 7;319(1):9-17. doi: 10.1016/s0009-8981(02)00015-3. PMID: 11922918] and MPO degranulation [Moore T, Wilcke J, Chilcoat C, Eyre P, Crisman M. Functional characterization of equine neutrophils in response to calcium ionophore A23187 and phorbol myristate acetate ex vivo. Vet Immunol Immunopathol. 1997 May;56(3-4):233-46. doi: 10.1016/s0165-2427(96)05750-9; Wright DG, Bralove DA, Gallin JI. The differential mobilization of human neutrophil granules. Effects of phorbol myristate acetate and ionophore A23187. The American Journal of Pathology. 1977 May;87(2):273-284]. In this case, the second process is longer and causes a lag phase. We added this note at the text of the article (page 8).
Page 9, Table 1: The experimental condition in which these assays have been preformed should be given here in the legend.
We added an appropriate description to Materials and Methods and to the table legend (pp 5 ,9-10).
Reviewer 2 Report
This manuscript reports on the potential use of the commercial due Celstine Blue (CB) as a ‘turn on’ fluorescent dye for the detection of HOCl, and the particular use of this dye to detect oxidant formation by activated neutrophils.
The studies reported are a follow up to the authors previous paper published in 2015, where they showed that this dye molecule could be used in an assay mixture containing taurine and iodide to detect HOCl and HOBr generated by myeloperoxidase. The current data are a limited extension of this previous work.
Matters for the attention of the authors:
11) A major concern is that the authors state that this dye is both sensitive and selective for HOCl. From my understanding of both the previous report from these authors and the chemistry involved, this is not the case. The authors have already shown that HOBr can oxidize CB, and although it is not shown, it is likely that a range of chloramines and bromamines will do likewise. The authors should therefore be much more circumspect as to how they describe the reactions under examination, and major changes should be made to the title, abstract and text to reflect this. It would also be very helpful if the authors could provide comparative data for a range of related species (pre-formed chloramines, bromamines, HOSCN, HOBr etc).
22) Related to point 1), it appears that H2O2 can also oxidize the probe (see Figure 1D) albeit to a lesser extent. However the data presented do not provide confidence that the probe could distinguish between HOCl and H2O2, when the former is present al low concentrations, and the latter at a much higher level (a likely in vivo scenario, due to the high reactivity of HOCl when compared to H2O2).
33) There are long sections of the Introduction that lack any references, and there are a significant number of statements with no supporting references.
44) The authors provide some data comparing CB to other probes, but mention that many of these are no available: It should be noted that R-19S is commercially available, and it should also be added to the Table comparing the available probes.
55) When compared to point 3) the inclusion of 16 references to various probes seems excessive. There are some review articles that have compared and contrasted some of these probes. The reader can just be referred to these.
66) There is no ethics data for the blood sample collection. This should be added.
77) Statistical analysis has been carried out using Student’s t-tests. This is not an appropriate methods when more than two data sets are compared (cf. Table 1).
88) How were the oxidant doses used in the studies reported in Figure 1D measured (assuming these are all the same concentration). If they were not quantified and equal, the comparison has much less validity.
99) The fluorescence detection of oxidized CB uses excitation at 430 nm. This is very close to the absorption bands for heme proteins (including MPO). It would therefore seem important to show that heme proteins, including MPO, do not interfere in the detection system via absorption quenching.
110) In figure 3B, ABAH is shown to provide near complete quenching of the CB signal, but in the flow cytometry experiments, the quenching is only 1.5-fold. Why this difference ? Why also does the probe show significant fluorescence (and ABAH inhibition) with unstimulated cells in these experiments compared to those reported earlier in the manuscript?
111) It is unclear why the oxidized CB localizes with the DNA in the NET experiments. CB is a relatively small molecule so why should this occur ?
112) Supplementary Table 1: ‘Plato time’ – do the authors mean time to attaining a plateau in absorbance /fluorescence ?
Author Response
We are grateful to both referees for a thorough study of our work and below we present answers to their questions.
Review â„–2
Matters for the attention of the authors:
1) A major concern is that the authors state that this dye is both sensitive and selective for HOCl. From my understanding of both the previous report from these authors and the chemistry involved, this is not the case. The authors have already shown that HOBr can oxidize CB, and although it is not shown, it is likely that a range of chloramines and bromamines will do likewise. The authors should therefore be much more circumspect as to how they describe the reactions under examination, and major changes should be made to the title, abstract and text to reflect this. It would also be very helpful if the authors could provide comparative data for a range of related species (pre-formed chloramines, bromamines, HOSCN, HOBr etc).
The main emphasis in the article is made on the possibility of using CB to detect HOCl in suspension of activated neutrophils. In our previous study Km for bromide oxidation by myeloperoxidase estimated as 2.1 mM [Sokolov A.V., Kostevich V.A., Zakharova E.T., Samygina V.R., Panasenko O.M., Vasilyev V.B. Interaction of ceruloplasmin with eosinophil peroxidase as compared to its interplay with myeloperoxidase: reciprocal effect on enzymatic properties. Free Radical Research, Vol. 46. 2015. P. 800-811. https://doi.org/10.3109/10715762.2015.1005615.], HOSCN can not oxidize CB [Sokolov A.V., Kostevich V.A., Kozlov S.O., Donskyi I.S., Vlasova I.I., Rudenko A.O., Zakharova E.T., Vasilyev V.B., Panasenko O.M. Kinetic method for assaying the halogenating activity of myeloperoxidase based on reaction of celestine blue B with taurine halogenamines. Free Radical Research, V. 46. 2015 P. 777-789. https://doi.org/10.3109/10715762.2015.1017478]. At physiological chlorine concentrations, the main product of MPO will be HOCl [Benjamin S. Rayner, Dominic T. Love, Clare L. Hawkins, Comparative reactivity of myeloperoxidase-derived oxidants with mammalian cells, Free Radical Biology and Medicine, Vol 71, 2014, P 240-255, https://doi.org/10.1016/j.freeradbiomed.2014.03.004; Arnhold J, Malle E. Halogenation Activity of Mammalian Heme Peroxidases. Antioxidants. 2022; 11(5):890. https://doi.org/10.3390/antiox11050890]. Indeed, the influence of halogenated proteins is possible. Therefore, in cell experiments, taurine was used, which is a common practice. It was also shown that, in a cell-free medium, CB can be used to detect HOCl without the participation of this mediator.
2) Related to point 1), it appears that H2O2 can also oxidize the probe (see Figure 1D) albeit to a lesser extent. However the data presented do not provide confidence that the probe could distinguish between HOCl and H2O2, when the former is present al low concentrations, and the latter at a much higher level (a likely in vivo scenario, due to the high reactivity of HOCl when compared to H2O2).
The study doesn’t aim at developing a probe for HOCl registration in vivo. We used up to 1 x 106 neutrophils/ml in experiments. Based on the literature, neutrophils (5 x 105 cells) are capable to produce up to 4 nM hydrogen peroxide within 15 min [Hoffstein ST, Gennaro DE, Manzi RM. Neutrophils may directly synthesize both H2O2 and O2- since surface stimuli induce their release in stimulus-specific ratios. Inflammation. 1985 Dec;9(4):425-37. doi: 10.1007/BF00916342]. In phagosomes, up to 2 µM of hydrogen peroxide is produced [Winterbourn CC, Hampton MB, Livesey JH, Kettle AJ. Modeling the reactions of superoxide and myeloperoxidase in the neutrophil phagosome: implications for microbial killing. J Biol Chem. 2006 Dec 29;281(52):39860-9. doi: 10.1074/jbc.M605898200]. In cell-free experiments, at concentrations up to 0.5 mM H2O2 had little effect on the intensity of CB fluorescence, which is also confirmed by the data of our previous article.
3) There are long sections of the Introduction that lack any references, and there are a significant number of statements with no supporting references.
We have added the missing links to the text of the article (page 2).
4) The authors provide some data comparing CB to other probes, but mention that many of these are no available: It should be noted that R-19S is commercially available, and it should also be added to the Table comparing the available probes.
The authors didn’t have information about the presence of R19-S in commercial sales. We have removed this comment. In the additional table, we have indicated only those probes that have been developed since 2016, as indicated in the caption to the table. We have added a more accurate presentation of this information in the text of the article (page 2).
5) When compared to point 3) the inclusion of 16 references to various probes seems excessive. There are some review articles that have compared and contrasted some of these probes. The reader can just be referred to these.
We have moved references to these probes to the supplementary materials. We have also added links to some review articles in this area in the body of the article (page 2).
6) There is no ethics data for the blood sample collection. This should be added.
Ethics data has been sent to the editor separately. Also we noted it in the article.
7) Statistical analysis has been carried out using Student’s t-tests. This is not an appropriate methods when more than two data sets are compared (cf. Table 1).
All data presented in the table are based on their control experiments. For example, the effect of PMA has been investigated in relation to its absence, fMLP and lectins + cyth b - relative to this activator without the presence of cyth b. We added an appropriate description to “Materials and Methods” (page 5) and to the table legend (pp 9-10).
8) How were the oxidant doses used in the studies reported in Figure 1D measured (assuming these are all the same concentration). If they were not quantified and equal, the comparison has much less validity.
This information is presented in the «Materials and Methods» section (pp 3-4). We also gave a decoding of oxidants in the legend to Figure 1. (page 7).
9) The fluorescence detection of oxidized CB uses excitation at 430 nm. This is very close to the absorption bands for heme proteins (including MPO). It would therefore seem important to show that heme proteins, including MPO, do not interfere in the detection system via absorption quenching.
In the section “Materials and Methods” describes that «To minimize neutrophil proteins contribution in the fluorescence assay, the excitation was performed at 460 nm».
10) In figure 3B, ABAH is shown to provide near complete quenching of the CB signal, but in the flow cytometry experiments, the quenching is only 1.5-fold. Why this difference ? Why also does the probe show significant fluorescence (and ABAH inhibition) with unstimulated cells in these experiments compared to those reported earlier in the manuscript?
The absence of complete inhibition of CB fluorescent response upon PMA activation of neutrophils may be due to the poor permeability of ABAH into cells. When MPO is degranulated into the extracellular medium, it is able to effectively inhibit MPO. Our results are similar to studies involving APF [Flemmig J, Zschaler J, Remmler J, Arnhold J. The fluorescein-derived dye aminophenyl fluorescein is a suitable tool to detect hypobromous acid (HOBr)-producing activity in eosinophils. J Biol Chem. 2012 Aug 10;287(33):27913-23. doi: 10.1074/jbc.M112.364299]. It should also be noted that the level of background activity of neutrophils and the degree of ABAH effect will also depend on the procedure of neutrophils isolation, donor feature, the time of exposure of neutrophils to the dye, inhibitor and activator, and etc.
11) It is unclear why the oxidized CB localizes with the DNA in the NET experiments. CB is a relatively small molecule so why should this occur ?
CB originally used in light microscopy for staining chromatin in acid medium, we cannot exclude than product of its oxidation interacted with DNA. NETs are DNA strands on which MPO, elastase and some other neutrophil proteins are found, which are necessary participants in the netosis process [Hamam HJ, Palaniyar N. Post-Translational Modifications in NETosis and NETs-Mediated Diseases. Biomolecules. 2019 Aug 14;9(8):369. doi: 10.3390/biom9080369]. This is a standard method for detecting NETs, when samples are stained not only for DNA, but also for MPO and NE (using AT) and show their colocalization. It also was shown than MPO is active during netosis [Parker, H., Albrett, A.M., Kettle, A.J. and Winterbourn, C.C. (2012), Myeloperoxidase associated with neutrophil extracellular traps is active and mediates bacterial killing in the presence of hydrogen peroxide. Journal of Leukocyte Biology, 91: 369-376. https://doi.org/10.1189/jlb.0711387]. So, our data only support this data. It doesn’t mean that there no CB anywhere else, that’s only stand that CB becomes oxidized in the region of DNA where MPO is concentrated.
12) Supplementary Table 1: ‘Plato time’ – do the authors mean time to attaining a plateau in absorbance /fluorescence ?
We compare parameters of fluorescent response. In view of this, the plateau time means the time to reach a plateau in the change in fluorescence intensity after HOCl addition. Appropriate transcript added to supplementary materials.
Round 2
Reviewer 1 Report
The manuscript can be published now.
Author Response
We are grateful to both referees for a thorough study of our work.
Reviewer 2 Report
Antioxidants review
This revised manuscript is significantly improved over the original version. However there are still a few minor issues that it would be beneficial for the authors to clarify / fix. Most of these concern the new revisions that the authors have included
1) The new title is an improvement, but this may give an impression to the reader that the probe also reacts with stable halogenated species such as 3-chlorotyrosine. It would be better to change this to something like: ‘…HOCl and reactive halogenated species’.
This should also be fixed in the abstract (line 29) and in the Discussion (line 472).
2) Line 49. This would be better written with ‘But’ changed to ‘However’ and ‘chlorine’ to ‘chloride’
3) Line 64: change to: amino acid, DNA/RNA bases or lipids
4) Line 95: change to: ….Tau-N-chloramine and becomes oxidized …..it change color to
5) Line 323: change to: two processes that need to happen……and MPO degranulation, which is a slow process.
6) Paragraph starting at line 463. It would be important to note here that the probe should only be employed in in vitro studies, with cells known to contain MPO, and that control experiments with MPO inhibitors (such as ABAH) need to be used to confirm that any changes in fluorescence are due to MPO-derived oxidants.
These statements are important as otherwise other workers may use it in other systems (or in vivo) where positive fluorescence signals may be detected from other (non-MPO derived oxidants) such as high concentrations of H2O2.
Author Response
1) The new title is an improvement, but this may give an impression to the reader that the probe also reacts with stable halogenated species such as 3-chlorotyrosine. It would be better to change this to something like: ‘…HOCl and reactive halogenated species’.
This should also be fixed in the abstract (line 29) and in the Discussion (line 472).
We have added “reactive halogenated species” in Title of manuscript, in abstract (line 29), and in Discussion (lines 425, 432, 439, 443, 448,461, 464, and 479=472 in previous numeration)
2) Line 49. This would be better written with ‘But’ changed to ‘However’ and ‘chlorine’ to ‘chloride’
We have added corresponding corrections in text (line 50).
3) Line 64: change to: amino acid, DNA/RNA bases or lipids
We have added corresponding correction in text (line 65).
4) Line 95: change to: ….Tau-N-chloramine and becomes oxidized …..it change color to
We have added corresponding corrections in text (line 96).
5) Line 323: change to: two processes that need to happen……and MPO degranulation, which is a slow process.
We have added corresponding corrections in text (lines 324-325).
6) Paragraph starting at line 463. It would be important to note here that the probe should only be employed in in vitro studies, with cells known to contain MPO, and that control experiments with MPO inhibitors (such as ABAH) need to be used to confirm that any changes in fluorescence are due to MPO-derived oxidants.
These statements are important as otherwise other workers may use it in other systems (or in vivo) where positive fluorescence signals may be detected from other (non-MPO derived oxidants) such as high concentrations of H2O2.
We have added corresponding corrections in text (lines 468-471): Important to note that CB should applied only for in vitro studies, with MPO-containing cells, and the control experiments with MPO inhibitors (such as ABAH) need to be used to confirm that any changes of fluorescence are due to MPO-derived oxidants (RHS).